# BMP-2/β-TCP Local Delivery for Bone Regeneration in MRONJ-Like Mouse Model

**DOI:** 10.3390/ijms21197028

**Published:** 2020-09-24

**Authors:** Akihiro Mikai, Mitsuaki Ono, Ikue Tosa, Ha Thi Thu Nguyen, Emilio Satoshi Hara, Shuji Nosho, Aya Kimura-Ono, Kumiko Nawachi, Takeshi Takarada, Takuo Kuboki, Toshitaka Oohashi

**Affiliations:** 1Department of Molecular Biology and Biochemistry, Okayama University Graduate School of Medicine, Dentistry and Pharmaceutical Sciences, Okayama 700-8558, Japan; a.mikai@s.okayama-u.ac.jp (A.M.); thuharhm@gmail.com (H.T.T.N.); de422038@s.okayama-u.ac.jp (S.N.); oohashi@cc.okayama-u.ac.jp (T.O.); 2Department of Oral Rehabilitation and Regenerative Medicine, Okayama University Graduate School of Medicine, Dentistry and Pharmaceutical Sciences, Okayama 700-8558, Japan; de421035@s.okayama-u.ac.jp (I.T.); a-kimura@md.okayama-u.ac.jp (A.K.-O.); nawachik@md.okayama-u.ac.jp (K.N.); kuboki@md.okayama-u.ac.jp (T.K.); 3Department of Biomaterials, Okayama University Graduate School of Medicine, Dentistry and Pharmaceutical Sciences, Okayama 700-8558, Japan; gmd421209@s.okayama-u.ac.jp; 4Center for Innovative Clinical Medicine, Okayama University Hospital, Okayama 700-8558, Japan; 5Department of Regenerative Science, Okayama University Graduate School of Medicine, Dentistry and Pharmaceutical Sciences, Okayama 700-8558, Japan; takarada@okayama-u.ac.jp

**Keywords:** BMP-2, MRONJ, bone regeneration

## Abstract

Medication-related osteonecrosis of the jaw (MRONJ) is a severe pathological condition associated mainly with the long-term administration of bone resorption inhibitors, which are known to induce suppression of osteoclast activity and bone remodeling. Bone Morphogenetic Protein (BMP)-2 is known to be a strong inducer of bone remodeling, by directly regulating osteoblast differentiation and osteoclast activity. This study aimed to evaluate the effects of BMP-2 adsorbed onto beta-tricalcium phosphate (β-TCP), which is an osteoinductive bioceramic material and allows space retention, on the prevention and treatment of MRONJ in mice. Tooth extraction was performed after 3 weeks of zoledronate (ZA) and cyclophosphamide (CY) administration. For prevention studies, BMP-2/β-TCP was transplanted immediately after tooth extraction, and the mice were administered ZA and CY for an additional 4 weeks. The results showed that while the tooth extraction socket was mainly filled with a sparse tissue in the control group, bone formation was observed at the apex of the tooth extraction socket and was filled with a dense connective tissue rich in cellular components in the BMP-2/β-TCP transplanted group. For treatment studies, BMP-2/β-TCP was transplanted 2 weeks after tooth extraction, and bone formation was followed up for the subsequent 4 weeks under ZA and CY suspension. The results showed that although the tooth extraction socket was mainly filled with soft tissue in the control group, transplantation of BMP-2/β-TCP could significantly accelerate bone formation, as shown by immunohistochemical analysis for osteopontin, and reduce the bone necrosis in tooth extraction sockets. These data suggest that the combination of BMP-2/β-TCP could become a suitable therapy for the management of MRONJ.

## 1. Introduction

In a super-aging society, the increasing number of patients with bone-resorption diseases (e.g., osteoporosis) has become an important issue of societal concern. Bone resorption inhibitors, such as bisphosphonates (BP), have been widely used for not only the treatment of osteoporosis but also suppression of bone metastatic cancer [1,2,3,4]. Rarely, however, individuals taking bone resorption inhibitors present with medication-related osteonecrosis of the jaw (MRONJ), particularly those who have suffered invasive dental treatments, such as tooth extraction. MRONJ is mainly characterized by incomplete healing of oral mucosa, jaw osteonecrosis, and necrotic jaw bone exposure with increased risk of infection, and induces significant burden and dysfunction to patients that affect the overall individual’s health and quality of life [5,6,7,8,9,10].

Although MRONJ was first reported in 2003 by Marx [11], the cause and pathophysiological mechanisms of MRONJ are still unclear, and the treatment method has not been clearly defined. It is known that bone resorption inhibitors cause an excessive suppression of osteoclast activities, resulting in the inactivation of bone remodeling and the increased susceptibility for oral bacterial infection [7,8,9]. Moreover, inhibition of angiogenesis is another major hypothesis in MRONJ pathophysiology since osteonecrosis itself is classically considered an interruption in vascular supply [7].

Bone Morphogenetic Protein (BMP)-2 is known to be a strong inducer of orthotopic and ectopic bone formation [12,13]. In the United States and Europe, the bone graft material (INFUSE^®^, Medtronic, USA), which uses collagen and human recombinant BMP-2 (rhBMP-2) derived from mammalian cells, has already been approved by the U.S. Food and Drug Administration (FDA) for clinical applications in cases requiring bone tissue regeneration in the orthopedic and dental fields [14,15,16,17]. BMP-2 is well known as a protein that strongly promotes osteoblast differentiation via BMP receptors [12,17,18]. On the other hand, BMP receptors have also been reported to be present in osteoclasts, and therefore, BMP-2 is also known to directly enhance osteoclast activities and overall bone remodeling [19,20]. Moreover, the BMPs are also known to be an important factor that controls angiogenesis and blood vessel maintenance [21]. Therefore, BMP-2 could have constructive effects on the reduced angiogenesis condition due to bone hardening, or even MRONJ-associated bone necrosis.

Nevertheless, BMP-2 has been considered to have only a moderate effect on the prevention or reconstruction of MRONJ-associated symptoms [22]. Moreover, Jung et al. have shown that the combination of parathyroid hormone (PTH) and BMP-2 helped to significantly improve bone formation compared with BMP-2 alone or a control group [23]. Additionally, a major limitation is regarding the fact that most studies have used collagen as a carrier, which does not allow space retention. Therefore, the utilization of bioceramics, which allow space-making, could be alternative materials for optimization of BMP-2 function in the prevention and treatment of MRONJ.

Hydroxyapatite (HAp) was one of the first developed bone replacement materials [24]. Despite its outstanding mechanical properties, the commercially available HAp shows no or poor resorption ability by osteoclasts [25,26]. On the other hand, the bioceramic beta-tricalcium phosphate (β-TCP) has been considered as an osteoconductive material that allows space retention and can be resorbed by osteoclasts, and therefore, be replaced by new bone. Moreover, β-TCP has also been approved by the U.S. FDA and is currently widely used for bone regeneration therapy in the fields of orthopedics and dentistry.

In our previous preclinical study, we developed a bone substitute material using Escherichia coli-derived rhBMP-2 (E-BMP-2) adsorbed in porous β-TCP as a carrier and succeeded in inducing bone formation in a porcine maxillary sinus floor elevation model [27]. Importantly, E-BMP-2/β-TCP has been recently approved by the Japanese Pharmaceuticals and Medical Devices Agency (PMDA) for application in bone augmentation before dental implant treatment in human clinical trials. However, it is unclear whether BMP-2/β-TCP could be also used both in the prevention and treatment of the symptoms related to MRONJ in humans. Therefore, the aim of this study was to evaluate the effect of BMP-2/β-TCP on bone formation in tooth extraction sockets in MRONJ prevention and treatment models in mice. The results demonstrated that local administration of BMP-2/β-TCP in the tooth extraction sockets significantly induced bone formation and reduced the bone necrosis in both MRONJ-like prevention and treatment models.

## 2. Results

### 2.1. Transplantation of BMP-2/β-TCP Could Not Accelerate the Bone Regeneration in the Tooth Extraction Socket during the Normal Wound Healing Process

First, in order to obtain a deeper understanding of the normal wound healing process of tooth extraction sockets, micro computed tomography (micro-CT) and histological analyses were performed at 1, 2 and 4 weeks after the extraction of maxillary first molars in mice. Micro-CT analysis showed that bone was not formed in the tooth extraction socket in the first week of healing, but appeared from the second week onward, showing an increase in density until the fourth week of healing (Figure 1A). Hematoxyline and Eosin (HE)-based histological analysis revealed immature woven bone formation in the tooth extraction socket at 1 week post-surgery (Figure 1B), with a bone-fill rate of 45.6% ± 11.5% (Figure 1E). Note that the tooth extraction socket was completely covered with epithelium. In the subsequent weeks, the tooth extraction socket was filled with new bone tissue, with a bone-fill rate of 87.1% ± 0.8% at 4 weeks post-surgery, and 91.1% ± 2.1% at 8 weeks post-surgery (data not shown). Notably, there was no significant difference in the total amount of bone regeneration from the fourth to the eighth week of healing. Additionally, these results indicate that the tooth extraction socket was almost completely replaced with the new bone after 4 weeks. Therefore, the subsequent experiments were followed up until the fourth week after tooth extraction.

Next, the effect of BMP-2/β-TCP on the normal wound healing process of tooth extraction socket was evaluated. Micro-CT analysis showed a radiopaque area corresponding to β-TCP in the tooth extraction socket after one and two weeks, but it was hardly distinguished at 4 weeks after transplantation (Figure 1C). Accordingly, histological analysis confirmed the presence of β-TCP in the tooth extraction socket after one to two weeks of transplantation (Figure 1D). Nevertheless, after four weeks, almost no β-TCP was observed in the tooth extraction socket, and the bone-fill rate reached 87.5% ± 3.7% (Figure 1E). Note that the area surrounding β-TCP was replaced by new bone after the initial week. Note also that while the bone regeneration rate of BMP-2/β-TCP was significantly low compared with normal healing in the initial week, possibly because of the slower resorption rate of β-TCP compared to the faster rate of bone formation, the bone regeneration rate in the BMP-2/β-TCP-transplanted group was not significantly different compared to that of the normal healing process of the tooth extraction socket either at 2 or 4 weeks post-surgery. From the above results, it was clarified that the transplantation of BMP-2/β-TCP could not accelerate bone regeneration during the normal wound healing process after 2 or 4 weeks of tooth extraction.

### 2.2. Transplantation of BMP-2/β-TCP Partially Induced Bone Formation in the Tooth Extraction Socket in a Mouse Model of MRONJ Prevention

The preventive effect of BMP-2/β-TCP on MRONJ onset was evaluated using a MRONJ-like model in mice. The experimental design is shown in Figure 2A.

Previously, we have reported that transplantation of β-TCP alone could not induce bone formation and that BMP-2 was required to reliably induce bone formation in a swine sinus lift model [27]. Additionally, since MRONJ is a severe clinical condition affecting bone remodeling, in this experiment, the experimental group was transplanted with the combination of BMP-2 and β-TCP. Micro-CT analysis showed radiopaque areas corresponding to the newly-formed bone around the highly radiopaque area corresponding to β-TCP in the tooth extraction socket of the BMP2/β-TCP transplanted group (Figure 2B). Note the absence of newly-formed bone in the control group.

Histological analysis revealed that the tooth extraction socket was mainly filled with a sparse tissue with few cellular components in the control group, indicating a disordered bone healing process (Figure 2C) On the other hand, in the BMP-2/β-TCP transplanted group, bone formation was observed at the apex of the tooth extraction socket, and was filled with a dense connective tissue rich in cellular components (Figure 2C). The bone-fill rate in the BMP-2/β-TCP transplanted group (32.5% ± 17.2%) was significantly higher than that in the control group (10.3% ± 7.9%) (Figure 2D). On the other hand, the number of empty lacunae of regenerated bone in tooth extraction sockets, which indicates the level of bone necrosis, was significantly decreased in the group transplanted with BMP-2/β-TCP (Figure 2E). In addition, Masson’s trichrome staining showed that the number of collagen fibers was significantly increased in the tooth extraction socket in the BMP-2/β-TCP transplanted group compared with the control group (Figure 2F,G).

### 2.3. Transplantation of BMP-2/β-TCP Strongly Induced Bone Formation in the Tooth Extraction Socket in a Mouse Model of MRONJ Treatment

To examine the therapeutic effect of BMP-2/β-TCP on MRONJ-associated bone loss, the tooth extraction socket was curetted two weeks after tooth extraction in a MRONJ-like model induced by administration of CY/ZA for 3 weeks, as shown in Figure 3A. Notably, during the two-week interval between tooth extraction and BMP-2/β-TCP transplantation, new bone tissue had not been formed, whilst only a sparse connective tissue was present inside the tooth extraction socket before curettage (data not shown).

To evaluate the bone formation ability of BMP-2/β-TCP, CY and ZA, which had been administered to induce the MRONJ-like lesion for a total of 5 weeks, were withdrawn after BMP-2/β-TCP transplantation into the curetted tooth extraction socket on week 5. Four weeks after BMP-2/β-TCP transplantation, micro-CT and histological analyses showed that, in the BMP-2/β-TCP transplanted group, the tooth extraction socket was almost completely filled with new bone, and the average bone-fill rate was 76.5% ± 11.5% (Figure 3B–D), which was close to that observed during normal healing of the tooth extraction socket (Figure 1E). In contrast, in the control group, most of the tooth extraction socket was filled with soft tissue and the average bone-fill rate was 25.2% ± 9.7% (Figure 3B–D). Note that the number of empty lacunae was significantly lower in the BMP-2/β-TCP transplanted group (Figure 3E).

Next, immunohistochemical analysis was performed for identification of the expression of osteopontin (OPN), a non-collagen protein secreted by osteoblasts, in the tooth extraction socket. The results showed that the area positive for OPN was markedly larger in the BMP-2/β-TCP transplantation group, compared with the control group (Figure 4A,B). Moreover, tartrate-resistant acid phosphatase (TRAP)-positive osteoclasts were also observed on the surface of the regenerated bone in the tooth extraction sockets in both the BMP-2/β-TCP transplantation and control groups, indicating a physiological bone turn-over (Appendix A). Furthermore, the expression of endomucin (EMCN), which is one of the vascular endothelial cell markers, was also analyzed. The results, however, revealed no significant difference in the positive area (percentage) of EMCN in the tooth extraction sockets between control and BMP-2/β-TCP-transplanted groups (Figure 4C,D).

## 3. Discussion

MRONJ is a critical clinical problem characterized by progressive necrosis of the jaw bone. In 2014, the American Association of Oral and Maxillofacial Surgeons (AAOMS) updated that MRONJ is defined as an exposed bone or bone that can be probed through an intraoral or extra-oral fistula(e) in the maxillofacial region, which does not heal within 8 weeks, and occurs in a patient who has received a bone-modifying agent or an angiogenic inhibitor, without a history of head and neck radiation [5,7]. It is understood that MRONJ is caused by two pharmacological agents: anti-resorptive including bisphosphonates and anti-angiogenic agents. However, the molecular and cellular mechanisms of MRONJ onset due to the administration of these drugs are yet unclear. Inhibition of osteoclast activity is indeed one of the main pathogeneses. Therefore, activation of bone remodeling could be one of the targets for the treatment of MRONJ. In the present study, we evaluated the preventive and therapeutic effects of BMP-2, which is well known to be a powerful osteoinductive cytokine and widely used for alveolar bone regeneration. The results demonstrated that the transplantation of BMP-2/β-TCP could accelerate the bone healing of tooth extraction sockets in both MRONJ prevention and treatment models.

Recently, cell transplantation therapies using stem cells or differentiated/mature cells have been performed not only for the treatment but also for prevention of diseases [28,29]. Kuroshima et al. have reported that systematic transplantation of adipose tissue-derived stromal vascular fraction (SVF) cells improved the bone necrotic lesions in the MRONJ-like mouse model [30]. Nevertheless, although cell transplantation therapy is expected to be a breakthrough for the prevention of MRONJ, it is usually costly and requires sophisticated management procedures.

Bone remodeling is a dynamic process involving resorption by osteoclasts and bone formation by osteoblasts. BMP-2 is known to directly regulate the osteogenesis of immature osteoblasts or progenitor cells by binding to the BMP receptor [18,31]. Moreover, BMP-2 is known to directly enhance osteoclast differentiation from progenitor cells [19,20]. Therefore, it is assumed that BMP-2 can activate bone remodeling by having direct effects on both osteoblasts and osteoclasts, and could be one major candidate cytokine for the prevention and treatment of MRONJ. Indeed, Brierly et al. have reported that the transplantation of hydrogels loaded with BMP-2 (BMP-2/hydrogel) in the tooth extraction socket prevented the onset of MRONJ in rats. However, the bone regeneration rate promoted by BMP-2/hydrogel was moderate, i.e., the bone volume/total volume in the tooth extraction sockets in the BMP-2/hydrogel group (69.1% ± 13.1%) increased only 1.36 times compared to the non-transplantation group (50.9% ± 8.8%). Moreover, a detailed analysis of necrotic bone with empty lacunae, which is one of the important parameters used to assess the severity of MRONJ, was not performed [22]. Jung et al. have reported a clinical preliminary study showing that only transplantation of BMP-2 absorbed with a collagen sponge (BMP-2/collagen) could not significantly induce bone formation at the MRONJ site, compared with the non-transplantation group, and that PTH administration would be necessary to stimulate the effect of BMP-2 in inducing bone formation [32]. These studies indicate that BMP-2 is an effective growth factor for MRONJ; however, its effectiveness is still suboptimal. Therefore, in this study, we utilized the combination of BMP-2/β-TCP because of the space-making capability of β-TCP, which allowed a 3.16-fold increase in the bone-fill rate of 32.5% ± 17.2% compared with the non-transplantation group (10.3% ± 7.9%). In the MRONJ treatment mouse model, transplantation of BMP-2/β-TCP after curettage in the tooth extraction socket could completely regenerate the bone with blood vessels (Figure 3), previously atrophied by the administration of angiogenesis suppressors, CY and ZA. These results suggest that the mechanical and physico-chemical properties of β-TCP that allow for space making and bone remodeling are of fundamental importance for bone regeneration in MRONJ, with superior effectiveness compared with soft materials, such as hydrogels and collagen sponges.

Angiogenesis is a key process in the wound healing of various tissues including bone. Therefore, the expression of EMCN, which is one of the vascular endothelial cell markers, was also analyzed in a mouse model of MRONJ treatment. Surprisingly, although EMCN positive vascular endothelial cells were observed at the healing site, the tooth extraction socket was not filled with newly-formed bone in the control group. On the other hand, the transplantation of BMP-2/β-TCP after curettage of the tooth extraction socket could promote a complete regeneration of the bone with blood vessels (Figure 3 and Figure 4). These data strongly indicate that not only activation of angiogenesis, but also activation of bone turn-over using drugs or growth factors, like BMP-2, are essential for the treatment of MRONJ patients.

Despite the promising bone-inducing effects of BMP-2, there are, however, important issues to be considered when delivering BMP-2 to patients. For instance, an increasing number of studies have reported complications, such as failure of bone regeneration and bone resorption associated with the clinical use of the BMP-2 [33]. Jeppsson et al. had already reported in 1996 that bone healing is inhibited by the treatment of BMP-2 in rabbits [34]; however, its mechanism remained unclear. Recently, our research group has reported that while the transplantation of BMP-2 induced bone formation in the bone marrow-scarce environment (tooth extraction sockets and calvaria), it conversely induced bone resorption in the bone marrow-abundant environments (mandible and femoral marrow) in vivo [35]. Additionally, we clarified that bone marrow cells inhibit BMP-2-induced osteoblast activity in vitro [35]. Therefore, special attention should be given to the characteristics of the host site where BMP-2 will be transplanted. In this present study, BMP-2/β-TCP was transplanted into the bone-marrow scarce tooth extracted sockets in both MRONJ-like prevention and treatment models, and therefore allowed an effective bone augmentation.

In summary, we for the first time demonstrated that the transplantation of BMP-2 adsorbed in β-TCP in the tooth extraction socket was effective for the prevention of the MRONJ onset, as well as for the treatment of MRONJ symptoms in an MRONJ-like mouse model. Although further confirmation using large animal models is required before application in the clinical setting, the results suggest that transplantation therapies of BMP-2 adsorbed in β-TCP could become a suitable treatment for MRONJ.

## 4. Materials and Methods

### 4.1. Materials

To prepare the E-rhBMP2/β-TCP complex, 1.5 mg of β-TCP (Superpore, particle size 0.6–1.0 mm, porosity 75%, HOYA, Tokyo, Japan) was immersed in 2.5 μL of 0.5 mM HCl containing 2.5 μg of BMP-2 (Osteopharma Inc., Osaka, Japan) and incubated for five minutes at room temperature. After incubation, almost all the solution was absorbed in the porous of β-TCP.

### 4.2. Animal Model

C57BL/6J female mice (8 week-old to 12 week-old females) were purchased from CLEA Japan Inc. (Tokyo, Japan). Recently, Kuroshima et al. [30,36] have reported that administration of CY and ZA can induce an MRONJ-like conditions such as inhibition of bone regeneration and angiogenesis, increased necrotic bone with empty lacuna in the tooth extraction sockets of mice. Therefore, in this study, MRONJ-like model mice were prepared according to their method [36]. Briefly, to create the MRONJ-like model, both zoledronate (ZA, 0.05 mg/kg, Zometa; Novartis, Stein, Switzerland) and cyclophosphamide (CY, 100 mg/kg, C7397; Sigma-Aldrich, St. Louis, MO, USA) were administrated subcutaneously and intraperitoneally, respectively, twice a week for 3 weeks. Three weeks after CY and ZA administration, the maxillary first molars were extracted under general anesthesia. After tooth extraction, MRONJ prevention and treatment models were designed.

In the case of the MRONJ prevention model, E-rhBMP2/β-TCP was transplanted into tooth extraction sockets immediately after tooth extraction. After transplantation, both CY and ZA were continuously administered twice a week for 4 weeks.

In the MRONJ treatment model, both CY and ZA were administered twice a week for 2 weeks after tooth extraction. The tooth extraction sockets were curetted by a dental probe and BMP-2/β-TCP was transplanted into the tooth extraction sockets. No medicine was administered in the subsequent four weeks. The control groups received no transplantation.

The animal experiment protocols used in this study (OKU-2019254, OKU-2020380) were approved by the Okayama University Animal Research Committee. All animals were handled according to the guidelines of the Okayama University Animal Research Committee.

### 4.3. Micro Computed Tomography (Micro-CT) Analysis

The collected samples were fixed with 4% paraformaldehyde (PFA; Merck, Kenilworth, NJ, USA) and analyzed by using a micro-computed tomography (micro-CT, SkyScan 1174, Bruker, Kontich, Belgium) as described previously [35]. The micro-CT parameters were as follows: 6.5 μm voxel size, 50 kVp, 800 µA, 1 mm aluminum filter, angular rotation step 0.7°, 180° scanning, 261 projections, and an exposure time of 4 s with a total scan duration of 37 min. Transmission images were reconstructed using SkyScan NReconc software (Bruker, Kontich, Belgium), and 2D images were obtained with SkyScan Data Viewer software (Bruker, Kontich, Belgium).

### 4.4. Histological Analysis

Fixed samples were decalcified with Morse solution (Shiyaku, Kyoto, Japan) for 1 week and embedded in paraffin. Sections of 5 μm were stained with Hematoxyline and Eosin (HE) and Masson’s trichrome according to standard protocols.

For immunohistochemical (IHC) analysis, frozen sections were prepared by using the Kawamoto’s film method, according to a previous report [37]. Briefly, samples fixed with 4% PFA were freeze-embedded in super cryoembedding medium (SECTION-LAB Co., Ltd., Hiroshima, Japan) and cut into widths of 5 μm with a tungsten carbide blade using an adhesive film. In addition, sections were blocked with 5% goat serum (Life technologies, Gaithersburg, MD, USA) for 60 min at room temperature (RT) and incubated with anti-osteopontin antibody (OPN: Immuno-Biological Laboratories, Gunma, Japan) and anti-endomucin antibody (EMCN: sc-65495, Santa Cruz Biotechnology, Dallas, TX, USA) overnight at 4 °C. Sections were washed and incubated with the secondary antibody, Alexa Fluor 488 goat anti-rabbit IgG or Alexa Fluor 647 goat anti-rat IgG (Life technologies, Gaithersburg, MD, USA) for 1 h at RT in a dark chamber. All images were taken with a BZ-710 fluorescence microscope (Keyence, Osaka, Japan) and quantitative analyses of bone-fill rate, the number of empty osteocyte lacunae, rate of collagen fiber into the remaining connective tissue and rate of OPN positive area and EMCN positive vessel area in the tooth extraction sockets were performed using the BZ analyzer (Keyence, Osaka, Japan).

## 5. Statistic Analysis

The results obtained from quantitative experiments were reported as the mean values  ±  SD. Statistical analyses were performed with two-way factorial ANOVA followed by Tukey’s multiple comparison tests and the unpaired Student’s *t*-test.

## Figures and Tables

**Figure 1 ijms-21-07028-f001:**
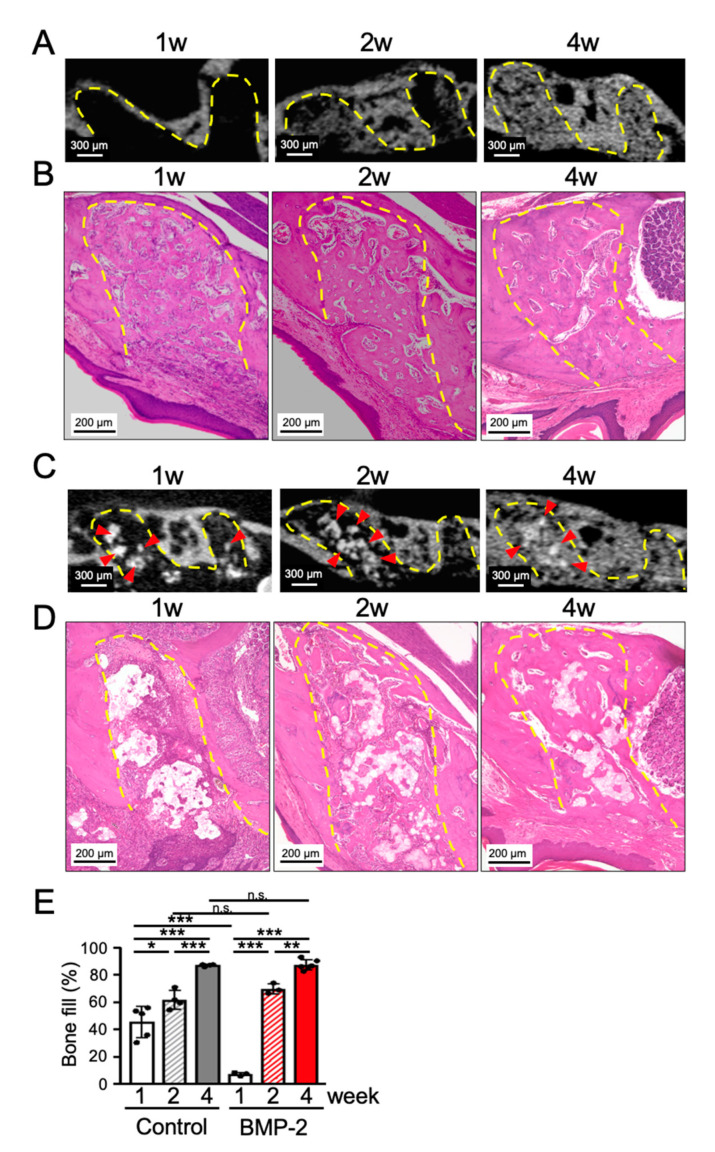
Transplantation of Bone Morphogenetic Protein (BMP)-2/beta-tricalcium phosphate (β-TCP) does not accelerate bone regeneration in the tooth extraction socket during the normal socket healing process. (**A**) Sagittal images of micro computed tomography micro-CT and (**B**) Hematoxyline and Eosin (HE)-stained sections after 1, 2 and 4 weeks of tooth extraction without any treatment. (**C**) Sagittal images of micro-CT and (**D**) HE-stained sections after 1, 2 and 4 weeks of BMP-2/β-TCP transplantation into the tooth extraction socket of the maxillary first molar. (**E**) The graph shows the quantitative analysis of the bone-fill rate in the tooth extraction socket. Bars represent the mean values and standard deviation (+/− SD) (*n* = 3–6). (* *p* < 0.05, ** *p* < 0.01, *** *p* < 0.001, ns: no significant difference. Two-way ANOVA/Tukey). Red arrowheads indicate the residual β-TCP. Yellow dashed lines indicate the tooth extraction socket.

**Figure 2 ijms-21-07028-f002:**
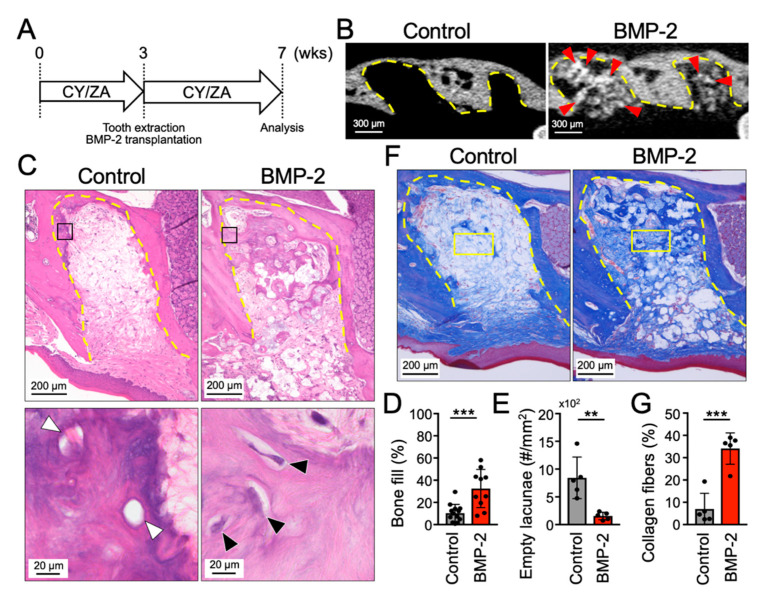
Transplantation of BMP-2/β-TCP partially induced bone formation in tooth extraction socket in a medication-related osteonecrosis of the jaw (MRONJ) prevention model. (**A**) Experimental design. BMP-2/β-TCP was transplanted into the extraction socket of the maxillary first molar after 3 weeks of CY/ZA administration in mice. CY/ZA administration was performed throughout the study period. ZA: zoledronate, CY: cyclophosphamide. (**B**) Sagittal images of micro-CT and (**C**) HE-stained sections, 4 weeks after transplantation. Lower panels are high magnification images of the squares in the upper HE-stained images. (**D**) The bone-fill rate in the tooth extraction socket and (**E**) the number of empty lacunae in the regenerated bone are shown in graphs, respectively. Bars represent the mean values and standard deviation (+/− SD) (*n* = 5–13). (** *p* < 0.01, *** *p* < 0.001. Student’s *t*-test). (**F**) Sagittal view of the tooth extraction socket stained with Masson’s trichrome, 4 weeks after transplantation. Yellow boxes indicate the total measured area for calculation of the ratio between the collagen fiber area and the total measured area. (**G**) The percentage area of the collagen fibers in the center of the tooth extraction socket is shown in the graph. Bars represent the mean values and standard deviation (+/− SD) (*n* = 5). (*** *p* < 0.001. Student’s *t*-test). Red arrowheads indicate the residual β-TCP. Black and white arrowheads indicate the osteocytes and empty lacunae, respectively. Yellow dashed lines indicate the tooth extraction socket.

**Figure 3 ijms-21-07028-f003:**
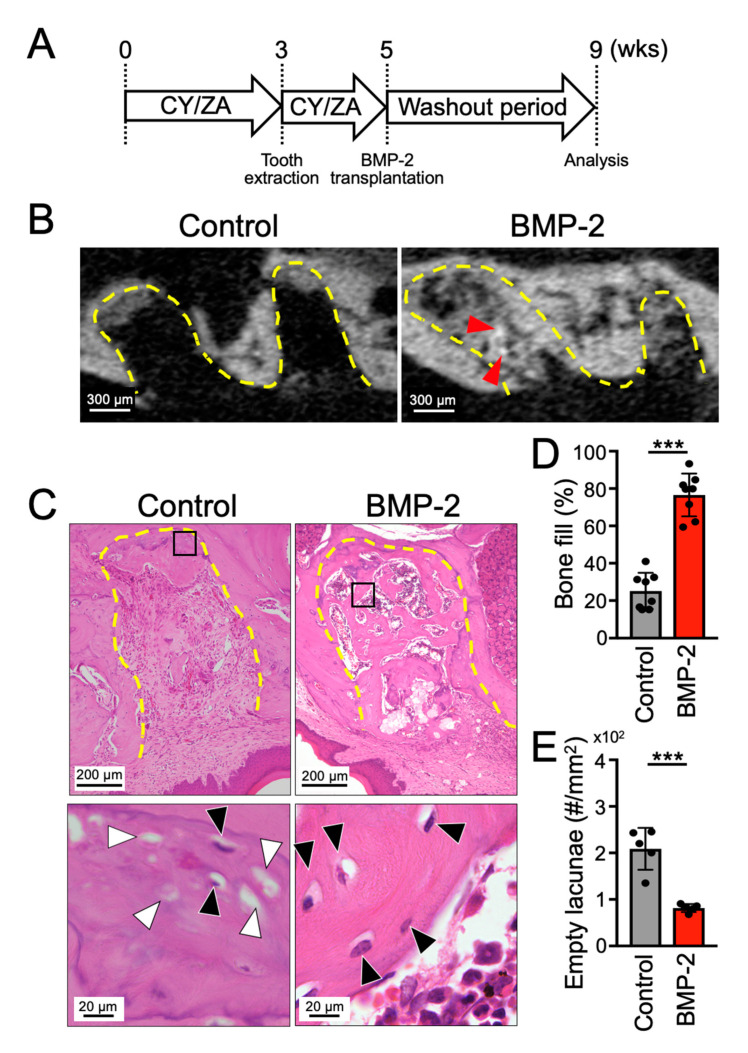
Transplantation of BMP-2/β-TCP strongly induced bone formation in the tooth extraction socket in an MRONJ-like treatment model. (**A**) Experimental design. The maxillary first molar of mice was extracted after 3 weeks of CY/ZA administration in mice. Two weeks after tooth extraction, BMP-2/β-TCP was transplanted into the tooth extraction socket under the cessation of CY/ZA administration. ZA: zoledronate, CY: cyclophosphamide. (**B**) Sagittal image of micro-CT and (**C**) HE-stained sections, 4 weeks after transplantation. Lower panels are high magnification images of the squares in the upper HE-stained images. (**D**) The bone-fill rate in the tooth extraction socket and (**E**) number of empty lacunae in regenerated bone are shown in graphs, respectively. Bars represent the mean values and standard deviation (+/− SD) (*n* = 5–8). (*** *p* < 0.001. Student’s *t*-test). Red arrowheads indicate the residual β-TCP. Black and white arrowheads indicate the osteocyte and empty lacunae, respectively. Yellow dashed lines indicate the tooth extraction socket.

**Figure 4 ijms-21-07028-f004:**
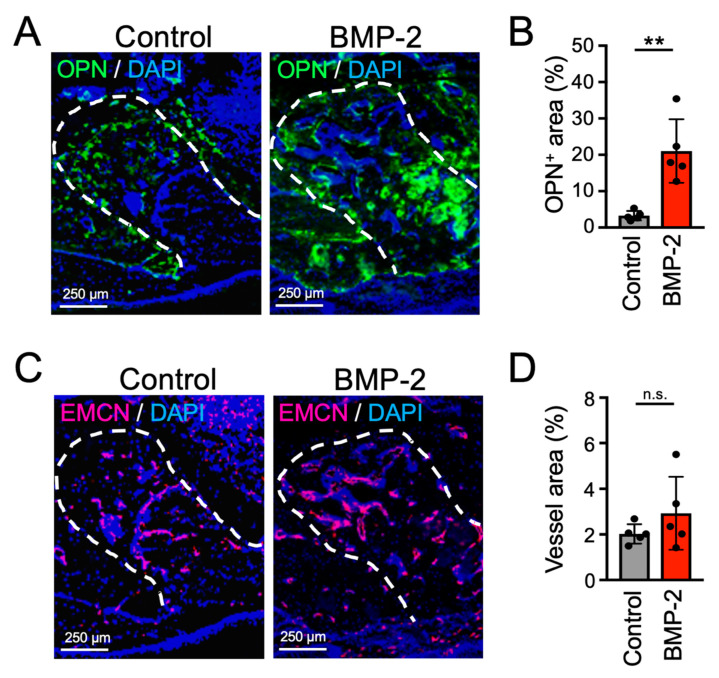
Immunohistochemical staining of OPN and EMCN in the tooth extraction socket in an MRONJ-like treatment model. (**A**) Immunohistochemical IHC staining images for OPN (green) and (**C**) EMCN (red) are shown, 4 weeks after transplantation. OPN: Osteopontin, EMCN: Endomucin. (**B**) The percentages of OPN positive area and (**D**) EMCN positive vessel area in the tooth extraction socket are shown in graphs. Bars represent the mean values and standard deviation (+/− SD) (*n* = 5). (** *p* < 0.01, ns: no significant difference, Student’s *t*-test). White dashed lines indicate the tooth extraction socket.

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
