# Peer review of "BMP-2/β-TCP Local Delivery for Bone Regeneration in MRONJ-Like Mouse Model"

_ijms, 2020, doi:10.3390/ijms21197028_

Round 1

Reviewer 1 Report

The manuscript entitled “BMP-2/β-TCP local delivery for bone regeneration in MRONJ-like mouse model” reports the effect of using BMP-2/β-TCP on bone regeneration in the extracted tooth socket of MRONJ mouse model. The study provides useful data about the subject, but the presentation needs significant improvement. In addition to language flaws that needs to be corrected, there are some other concerns that need to be addressed before further consideration. The following comments may help the authors improve their work:  

  • More key information such as results in the abstract compared to background information
  • Please add the novelty of the work to introduction.
  • Please give extra care in preparing the figure captions e.g. fig 1: panel (A) has been mentioned twice. Please make the description of each panel consistent by mentioning the Panel ID (letter) either in the beginning or at the end. Also please emphasize in the caption that panels A-C are related to the socket without any treatment (you can call it control group maybe). Very similar things can be seen in other figure captions. Also please check all the figure captions for accuracy of description. For example fig3, you have stated: “White and black arrow heads indicate the osteocyte and empty lacunae, respectively”, what I see in the figure is opposite.
  • The authors state: “Note also that the bone regeneration rate of BMP-2/β-TCP was equivalent to the normal healing process of the tooth extracted wound at four weeks. I don’t think the statement is precise enough. Even tough at 4 and 8 weeks, the bone formation is very similar, there is a significant difference at week 1 that should be discussed. The reason might be the slower resorption rate of beta-TCP compared to the faster rate of bone formation. Also, I think it would be very helpful to combine panels C and F and conduct a statistical analysis between control (no treatment) and beta-TCP/BMP and according to that analysis you might be able to correct your statement as: at weeks 2 and 4, there was no significant difference between control and TCP/BMP demonstrating …
  • Is there any reason the authors have used the term MRONJ-like model? It is not a MRONJ model?
  • In the sentence “The bone-fill-rate in the BMP-2/β-TCP transplanted group (32.5%) was significantly higher than that in control group (Fig. 2D)”, please use mean ± SD to present the data and also present the value (mean ± SD) for the control group in order to enable comparison.
  • In section 2-3, authors mention “Next, CY and ZA, which had been administered to induce the MRONJ-like lesion, were withdrawn, and BMP-2/β-TCP was transplanted into the curetted tooth extraction socket”. Please be more specific about the time points in the text, for example: CY and ZA, which had been administered to induce the MRONJ-like lesion for X weeks, were withdrawn Y, and BMP-2/β-TCP was transplanted on week Z
  • The authors have referred to Fig 3B-D in the following sentence in section 2.3: “Four weeks after transplantation, micro-CT and histological analyses showed that in the BMP-2/β-TCP transplanted group, the tooth extraction socket was almost completely filled with new bone, and the average of bone-fill-rate was 76.5%, similar to the ration observed during normal healing of tooth extraction socket (Fig. 3B-D). which is inaccurate; fig 3 is not about normal healing. Even if you mean fig 1, the panels B-D are inaccurate. All the refers to figures in the manuscript must be double checked. If you wanted to refer to the MRONJ model and data in fig 3, it is a confusing way of doing that. instead you should write: “Four weeks after transplantation, micro-CT and histological analyses showed that in the BMP-2/β-TCP transplanted group, the tooth extraction socket was almost completely filled with new bone, and the average of bone-fill-rate was 76.5% (Fig. 3B-D), similar to the ration observed during normal healing of tooth extraction socket”. Should refer to the figure at proper place in the text.
  • Please add complications of using BMP in clinic to the discussion
  • What is the advantage of authors work over others like Brierly et al. and why? Is it partially but significantly induced bone formation? Why do think beta-TCP is a better carrier and leads to enhanced bone formation? Please discuss.
  • In the materials and methods section, more details need to be provided for micro-CT scan and reconstruction such as voltage, object to detector distance, rotation angle, reconstruction software etc.
  • Details should be provided about the preparation procedure of beta-TCP/BMP2, such as buffer used, incubation time, specs for beta-TCP such as granule size, porosity if available. Do you think BMP is adsorbed and then released from TCP?

Author Response

Reviewer 1
The manuscript entitled “BMP-2/β-TCP local delivery for bone regeneration in MRONJ-like mouse model” reports the effect of using BMP-2/β-TCP on bone regeneration in the extracted tooth socket of MRONJ mouse model. The study provides useful data about the subject, but the presentation needs significant improvement. In addition to language flaws that needs to be corrected, there are some other concerns that need to be addressed before further consideration. The following comments may help the authors improve their work:  

Comment #1: More key information such as results in the abstract compared to background information. 

Response #1: The authors thank the reviewer’s comment, and have modified the abstract accordingly.

Comment #2: Please add the novelty of the work to introduction.

Response #2: The authors thank the reviewer’s comment, and have included additional sentences in the Introduction section to clarify the novelty of this study (pages 2 lines 70-80, 86-87).

Briefly, the novel points in this study are:

- Utilization of the bioceramic β-TCP as a carrier for BMP-2 in both prevention and treatment of MRONJ.

- Presentation of concrete dataset showing bone formation by BMP-2/β-TCP in both prevention and treatment MRONJ models by means of micro-CT as well as histological (HE, Masson’s trichrome staining) and immunohistochemical (osteopontin for osteoblast-formed bone, endomucin for endothelial cells) analyses.

- Evaluation of the effects of BMP-2/β-TCP in the normal healing process of tooth extraction socket as well as in MRONJ-induced models.

Comment #3: Please give extra care in preparing the figure captions e.g. fig 1: panel (A) has been mentioned twice. Please make the description of each panel consistent by mentioning the Panel ID (letter) either in the beginning or at the end. Also please emphasize in the caption that panels A-C are related to the socket without any treatment (you can call it control group maybe). Very similar things can be seen in other figure captions. Also please check all the figure captions for accuracy of description. For example fig3, you have stated: “White and black arrow heads indicate the osteocyte and empty lacunae, respectively”, what I see in the figure is opposite.

Response #3: The authors greatly thank the detailed observation made by the reviewer, and have edited the figure legends carefully.

Comment #4: The authors state: “Note also that the bone regeneration rate of BMP-2/β-TCP was equivalent to the normal healing process of the tooth extracted wound at four weeks. I don’t think the statement is precise enough. Even tough at 4 and 8 weeks, the bone formation is very similar, there is a significant difference at week 1 that should be discussed. The reason might be the slower resorption rate of beta-TCP compared to the faster rate of bone formation. Also, I think it would be very helpful to combine panels C and F and conduct a statistical analysis between control (no treatment) and beta-TCP/BMP and according to that analysis you might be able to correct your statement as: at weeks 2 and 4, there was no significant difference between control and TCP/BMP demonstrating …

Response #4: The authors greatly thank the reviewer’s critical comment, and have edited the manuscript and Figure 1 accordingly (Figure 1 and pages 4 lines 126-132).

Comment #5: Is there any reason the authors have used the term MRONJ-like model? It is not a MRONJ model?

Response #5: The authors thank the reviewer’s question. The incidence of MRONJ in humans is 0.01-1%, and it is very challenging to perform the experiments in mice with that incidence. Therefore, the mice are administered approximately 10 times more drug per weight than in humans to induce the MRONJ-like lesion. Therefore, many articles use the words “MRONJ-like model”, instead of “MRONJ model”.

Comment #6: In the sentence “The bone-fill-rate in the BMP-2/β-TCP transplanted group (32.5%) was significantly higher than that in control group (Fig. 2D)”, please use mean ± SD to present the data and also present the value (mean ± SD) for the control group in order to enable comparison.

Response #6: The authors thank the reviewer’s comment, and have added the information of SD in the manuscript (pages 5 lines 164-165).

Comment #7: In section 2-3, authors mention “Next, CY and ZA, which had been administered to induce the MRONJ-like lesion, were withdrawn, and BMP-2/β-TCP was transplanted into the curetted tooth extraction socket”. Please be more specific about the time points in the text, for example: CY and ZA, which had been administered to induce the MRONJ-like lesion for X weeks, were withdrawn Y, and BMP-2/β-TCP was transplanted on week Z.

Response #7: The authors thank the reviewer’s comment, and have edited the manuscript according to the reviewer’s suggestion in section 2-3 (pages 6-7 lines 173-193).

Comment #8: The authors have referred to Fig 3B-D in the following sentence in section 2.3: “Four weeks after transplantation, micro-CT and histological analyses showed that in the BMP-2/β-TCP transplanted group, the tooth extraction socket was almost completely filled with new bone, and the average of bone-fill-rate was 76.5%, similar to the ration observed during normal healing of tooth extraction socket (Fig. 3B-D). which is inaccurate; fig 3 is not about normal healing. Even if you mean fig 1, the panels B-D are inaccurate. All the refers to figures in the manuscript must be double checked. If you wanted to refer to the MRONJ model and data in fig 3, it is a confusing way of doing that. instead you should write: “Four weeks after transplantation, micro-CT and histological analyses showed that in the BMP-2/β-TCP transplanted group, the tooth extraction socket was almost completely filled with new bone, and the average of bone-fill-rate was 76.5% (Fig. 3B-D), similar to the ration observed during normal healing of tooth extraction socket”. Should refer to the figure at proper place in the text.

Response #8: The authors greatly thank the detailed observation made by the reviewer, and have edited the manuscript accordingly (Pages 7 lines 193-197).

Comment #9: Please add complications of using BMP in clinic to the discussion

Response #9: The authors thank the reviewer’s comment, and have discussed the complication of BMP-2 in the discussion section (pages 9 lines 285-297).

Comment #10: What is the advantage of authors work over others like Brierly et al. and why? Is it partially but significantly induced bone formation? Why do think beta-TCP is a better carrier and leads to enhanced bone formation? Please discuss.

Response #10: The authors thank the reviewer’s comment, and have discussed advantages of our research in the discussion section (pages 8-9 lines 245-265).

Comment #11: In the materials and methods section, more details need to be provided for micro-CT scan and reconstruction such as voltage, object to detector distance, rotation angle, reconstruction software etc.

Response #11: The authors thank the reviewer’s comment, and have added the detail information of micro-CT analysis in the materials and methods section (pages 10 lines 323-327).

Comment #12: Details should be provided about the preparation procedure of beta-TCP/BMP2, such as buffer used, incubation time, specs for beta-TCP such as granule size, porosity if available. Do you think BMP is adsorbed and then released from TCP?

Response #12: The authors thank the reviewer’s comment, and have added the detail information of preparation procedure of β-TCP in the materials and methods section (pages 9 lines 295-298). Yes, the authors believe that BMP-2 is adsorbed and then released from beta-TCP, as reported previously (Ono, Cells Tissues Organs, 2014).

Reviewer 2 Report

This paper was well written on the subject of bMP2/TCP could become a suitable therapeutic for MRONJ. The manuscript can be accepted for publication after revision based on the comments provided below.

- The normal healing process was displayed after 8 weeks, whereas all results were observed 4 weeks after implantation in MRONJ models. Results for Figures 2, 3 and 4 also need to display the 8-week results.
- In addition, when the MRONJ-like mouse model is produced, it is necessary to display the actual necrosis in a little more detail. Also, it is necessary to explain how the area where necrosis occurred in the control is filled with a soft tissue.
- An enlarged image requires a mark on which part it zoomed in.
- In the enlarged image of Figure 3c, inflammatory cells are observed when looking at the bottom right. It raises doubts about whether BMP2/TCP plays a role as a treatment in the MRONJ model.

Author Response

Reviewer 2

This paper was well written on the subject of BMP2/TCP could become a suitable therapeutic for MRONJ. The manuscript can be accepted for publication after revision based on the comments provided below.

Comment #1: The normal healing process was displayed after 8 weeks, whereas all results were observed 4 weeks after implantation in MRONJ models. Results for Figures 2, 3 and 4 also need to display the 8-week results.

Response #1: The authors thank the reviewer’s comment. As shown in the old Fig. 1, there was no change in the total amount of bone regeneration from the fourth to the eighth week of healing, therefore, the subsequent experiments were followed up until the fourth week after tooth extraction. This information was added in the revised manuscript (pages 3, lines 104-108), and the data of normal bone healing at 8 weeks post-tooth extraction has been removed from the new Fig. 1.

   Additionally, reviewer #1 suggested to combine the panels C and F in the old Figure 1 and conduct a statistical analysis between the control group and beta-TCP/BMP-2 group at 1, 2 and 4 weeks post-transplantation. This was another reason to remove the data of normal bone healing at 8 weeks post-tooth extraction.

Comment #2: In addition, when the MRONJ-like mouse model is produced, it is necessary to display the actual necrosis in a little more detail. Also, it is necessary to explain how the area where necrosis occurred in the control is filled with a soft tissue.

Response #2: The authors thank the reviewer’s comment, and have edited the manuscript in the materials and methods section, according to the reviewer’s suggestion (pages 9 lines 301-303).

Comment #3: An enlarged image requires a mark on which part it zoomed in.

Response #3: The authors thank the reviewer’s comment, and edited Figure 2C and 3C, accordingly.

Comment #4: In the enlarged image of Figure 3c, inflammatory cells are observed when looking at the bottom right. It raises doubts about whether BMP2/TCP plays a role as a treatment in the MRONJ model.

Response #4: The authors greatly thank the reviewer’s comment. As the reviewer mentioned, inflammatory cells can be observed at the regeneration site. In fact, BMP-2 is known to induce mild inflammation in the initial days after transplantation (Kwang-Bok Lee et al., Journal of Orthopaedic Research, 2012). After that, BMP-2 strongly induced bone formation, without significant signs of inflammation. Additionally and alternatively, the authors think that the cells shown in Fig. 3C are blood cells constituting a bone marrow microenvironment at the regenerated site, for the following reasons:

  1. Authors have transplanted the BMP-2/β-TCP into the dorsal region of WT and MRONJ-like model mice to evaluate the ectopic bone formation. We could observe the cells with similar morphology of that of bone marrow cells in BMP-2 induced-ectopic bone formation, and that of the cells observed in Figure 3C.
  2. We have also performed the flow cytometric analysis of the cells present in the bone formed by BMP-2/β-TCP transplantation into the dorsal region of WT and MRONJ-like model, and identified the cells listed below, which include hematopoietic stem and progenitor cells.

1- HSC: LinSca-1+c-Kit+CD150+CD48−, MPP: LinSca-1+c-Kit+CD150CD48

2- CLP:  LinSca-1lowc-KitlowIL-7Ra+CD135+, CMP: LinSca-1c-Kit+CD16/32CD34+

3- MEP: LinSca-1c-Kit+CD16/32CD34−, GMP: LinSca-1c-Kit+CD16/32+CD34+

4- Pre-pro-B cell: B220+CD43+CD24IgM−, Pro-B cell: B220+CD43+CD24+IgM

5- Pre-B cell: B220+CD43CD24+IgM−, T cell: CD3+, NK cell: CD3-NK1.1+

6- Erythroid cell: Ter119+, Myeloid cell: CD11b+Gr-1+

These data strongly indicate that BMP-2 can generate bone marrow. Therefore, we are thinking that the cells observed in Figure 3C are bone marrow cells. The authors are preparing a new manuscript with this dataset, and would prefer to not include it in this present paper. The authors would appreciate the reviewer’s understanding on this issue.

Round 2

Reviewer 1 Report

The authors have addressed my comments.